# Suprabasin—A Review

**DOI:** 10.3390/genes12010108

**Published:** 2021-01-18

**Authors:** Miroslav Pribyl, Zdenek Hodny, Iva Kubikova

**Affiliations:** Laboratory of Genome Integrity, Institute of Molecular Genetics of the Czech Academy of Sciences, Videnska 1083, 14220 Prague, Czech Republic; hodny@img.cas.cz (Z.H.); iva.kubikova@img.cas.cz (I.K.)

**Keywords:** suprabasin, cancer resistance, immunity, cancer stem-like cells, interferon, MAPK signalling

## Abstract

Among the ~22,000 human genes, very few remain that have unknown functions. One such example is suprabasin (*SBSN*). Originally described as a component of the cornified envelope, the function of stratified epithelia-expressed *SBSN* is unknown. Both the lack of knowledge about the gene role under physiological conditions and the emerging link of *SBSN* to various human diseases, including cancer, attract research interest. The association of *SBSN* expression with poor prognosis of patients suffering from oesophageal carcinoma, glioblastoma multiforme, and myelodysplastic syndromes suggests that *SBSN* may play a role in human tumourigenesis. Three *SBSN* isoforms code for the secreted proteins with putative function as signalling molecules, yet with poorly described effects. In this first review about *SBSN*, we summarised the current knowledge accumulated since its original description, and we discuss the potential mechanisms and roles of *SBSN* in both physiology and pathology.

## 1. Introduction

Since its original description in human and mouse keratinocyte differentiation [1], suprabasin (*SBSN*) has been associated with multiple diseases, including cancer. SBSN isoforms are putative signalling molecules inducing cellular signalling (AKT, WNT/β-catenin, and/or p38MAPK signalling) and various cellular processes, such as migration, proliferation, neovascularization, therapy-, apoptosis- and immune-resistance. Therefore, *SBSN* is considered an oncogene and is a proposed biomarker in a couple of diseases, lung carcinoma and myelodysplastic syndromes (MDS). The apparent need for a deeper understating of the nature and the function of *SBSN* prompted us to compile all current knowledge of *SBSN* together with suggestions for future research work.

## 2. SBSN Gene Organization

*SBSN* gene is located on human chromosome 19 (chr 19: 35,523,367–35,528,351 reverse strand; GRCh38:CM000681.2; band 19q13.1) close to other keratinocyte-differentiation associated genes dermokine-α/β and *KDAP* [1,2]. In mice it corresponds to chromosome 7 (7:30,751,471–30,756,134 forward strand; GRCm38:CM001000.2; band 7B2–7B3). The *SBSN* gene is a part of a coordinately expressed new stratified epithelium-related gene cluster, tentatively named stratified epithelium secreted peptides complex, SSC [2]. *SBSN* was studied mostly in human and mice; however, other mammals possess *SBSN* homologs. Predicted SBSN peptide encoded in the genome of *Gorilla gorilla gorilla* possesses 97.6% amino acid sequence similarity to the human *SBSN-1*. Relatively high amino acid sequence identity (58.9%) between mouse and human orthologs of the largest SBSN isoform (SBSN isoform 1; SBSN-1) suggests strong gene integrity and conserved function among the species. Paralogs of *SBSN* were not defined, and no genes with confidently high sequence identities were identified. The Ensembl database [3] (Ensembl Genome Browser version 101, accessed on 21 August 2020) enlists 106 *SBSN* orthologs (26/26 primates; 30/32 rodents and related; 39/45 Laurasiatheria; 0/19 Sauropsida; 2/86 fish, and nine Monotremata and Marsupialia; Figure 1) with 96 orthologs having Gene Order Conservation Score 100 (identical four closest genes), implicating true orthology. Note, the putative *SBSN* fish genes show Gene Order Conservation Scores of 0 and 2.1% and 8.6% sequence identity to the human ortholog, respectively. Altogether, these observations suggest that *SBSN* is likely a mammalian-specific orphan gene.

Human *SBSN* gene consists of five exons and four introns (Figure 2a). Human *SBSN* mRNA can be alternatively spliced producing three known isoforms *SBSN-1* (transcript length: 1946 bp; ENST00000452271.7), *SBSN-2* (957 bp; ENST00000518157.1) and *SBSN-3* (593 bp; ENST00000588674.5). Importantly, *SBSN-2* represents a fully spliced isoform [5], whereas *SBSN-1* contains an in-frame retention of the first intron. *SBSN-3* is also a fully spliced isoform but lacks exon 2. Strikingly, the mouse (*Mus musculus*) isoform corresponding to human *SBSN-2* has not been identified. This may restrict the applicability of a mouse model for functional studies of *SBSN* isoforms. Interestingly, a sequence with high sequence identity (81.4%) to human exon two is present within the murine *Sbsn-1* exon one, and the major difference between the human intron one exon two junction and homologous murine sequence (CACGAGGCCGGG vs. AACCAGGGTCAA) implies that in mice, the splicing site was likely not established, or was lost. The murine putative exon two is spliced together with the entire exon one in murine *Sbsn-2*, hence resembling human isoform three, based upon amino acids sequence identity analysis (44.5% for human *SBSN-2* vs. murine *Sbsn-2*, and 65.5% for human *SBSN-3* vs. murine *Sbsn-2*). Multiple other rodents lack homolog corresponding to the human *SBSN-2* isoform, and interestingly, several primates (e.g., *Macaca mulatta*, *Pan paniscus* and *Microcebus murinus*) lack putative *SBSN-2* as well. However, this may be due to the absence of its identification, rather than sequence deviation.

## 3. SBSN Protein Structure

The UniProtKB database [6] (The UniProt release 2020_4, accessed 21 August 2020) refers to three human *SBSN* isoforms (Figure 2b). SBSN-1 (Q6UWP8-1; 590 aa, predicted mass 60.541 Da) and SBSN-2 (Q6UWP8-2; 247 aa, predicted mass 25.335 Da) are well defined. However, SBSN-3 (K7ESC4) is described as a 149 aa long peptide with a predicted mass of 15.318 Da, which is incorrect due to the lacking description of putative N-terminal signal peptide (i.e., missing N-terminal sequence MHLARLVGSCSLLLLLGALS; see also sequence alignment in Figure 2a) in the database. In fact, the SBSN-3 coding sequence is 507 nts long and translates into a 169 aa long peptide. All SBSN isoforms possess a putative N-terminal signal peptide (aa 1–25) [2,7] granting SBSN secretory nature and extracellular localization.

Distinctly of the other two isoforms, SBSN-1 is alanine-(14%), glycine-(20%), and histidine-(10%) rich due to the presence of short tandem Glycine-x-Histidine-Histidine repeats (GxHH repeats, x stands for any classical amino acid) encoded by the retained intron one. Note, structural proteins often contain compositional biases of similar characteristics to SBSN-1 Ala-/Gly-/His-rich domain (aa 26—524), and based upon these observations, SBSN was proposed as a component of the cornified envelope (CE), the essential structure of corneocytes responsible for the skin protective function [1,7]. Notably, no similar sequence encoding SBSN-1 Ala-/Gly-/His-rich domain is present in the human genome, indicating an evolutionarily unique and conserved role. This is further supported by a comparison of *SBSN* among species, showing remarkable conservation of the Ala-/Gly-/His-rich domain, as well as retention of the first intron encoding the domain. Human SBSN-2 and SBSN-3 lack most of the Ala-/Gly-/His-rich domains due to intron one splicing; however, a short repeat is located at the 3′ end of exon one. Similarly to Ala-/Gly-/His-rich domain of SBSN-1, the C-terminal sequence of all isoforms is of unknown function. As mentioned, SBSN-3 lacks exon two, but the remaining exons are present in all human SBSN isoforms.

The mouse homolog of *SBSN* possesses similar features. The N-terminal signal peptide was predicted (aa 1–23), and the compositional bias of Ala-/Gly-/His-rich domain (aa 97—478) in SBSN-1 (Q8CIT9; 700 aa; predicted mass 72.334 Da) is also present in mouse homolog. The current description of other murine Sbsn isoforms is rather confusing. Originally, murine Sbsn-1 isoform was identified as a 700 aa long peptide [1]. A shorter isoform, lacking the longest isoform-specific internal repeats, is a putative Sbsn-2 (Q8CIT9-3; 164 aa; predicted mass 16.967 Da), which corresponds to human SBSN isoform-3 according to sequence analysis. Besides these two isoforms, Ensembl (version 101) identifies two other protein-coding isoforms supported by one or no EST, respectively. The UniProtKB database refers to six mouse Sbsn isoforms, including Sbsn-1 and Sbsn-2, the other isoforms are either duplications of Sbsn-1 or Sbsn-2, or unreviewed isoforms.

Secreted murine Sbsn-2 was detected using overexpression experiments [7], showing a mobility shift on immunoblots which indicates a post-translational modification. Indeed, murine Sbsn-1 is a substrate to tissue transglutaminase (Tgm) 2 and epithelial Tgm3 resulting in intermolecular crosslinking, and this represents the only verified post-translational modification in vitro. Its physiological relevance is still undetected. Citrullination of Sbsn was observed in the blood proteome of Parkinson’s disease in a rat model of pre-motor Parkinson’s disease [8]. Additionally, glycosylation of SBSN-1 was predicted [7] and indeed, using mass spectrometry approach, threonine 59 of human SBSN was identified to undergo *O*-glycosylation in various human cell lines [9]. Three protein kinase C (PKC) phosphorylation sites in C-terminus and two casein kinase II phosphorylation sites were predicted together with 73 potential N-myristoylation sites in the mouse homolog [1]; however, these predictions require experimental validation.

Three-dimensional structure of SBSN is still unknown. We performed Phyre2-based structure prediction [10] of all three human SBSN isoforms. In the case of SBSN-1, the predicted model predominantly consisted of disordered regions. Crystal structure of d337a mutant of *Pseudomonas* sp. mis38 lipase (PDB: 2ZJ6) served as a template for structure prediction with 12% sequence identity and 99.9% prediction confidence. Collagen α chain was among other possible templates with slightly lower confidence levels. The methyl-accepting chemotaxis protein of *Escherichia coli* (PDB: 1QU7), showing 13% sequence identity with SBSN-2, was used as a template for the structure prediction of the isoform with 98.5% confidence. Interestingly, the methyl-accepting chemotaxis protein of *Thermotoga maritima* (PDB: 2CH7) was suggested as a second template with 98.4% confidence. SBSN-3 isoform model was predicted upon the structure of human micelle-bound α-synuclein (PDB: 1XQ8), which showed 18% sequence identity.

## 4. Regulation of SBSN Expression

*SBSN* expression is tightly associated with stratified epithelia, but other expression sites have also been defined. Mechanisms of *SBSN* transcriptional control are not clearly described. We utilized the JASPAR algorithm [11] to predict binding sites of transcription factors in human (Appendix A) and mouse (Appendix A) *SBSN* proximal promoter (2 kbp upstream) and downstream coding region. Additionally, we verified putative binding sites. Based on these results, we established a model of human *SBSN* promoter (Figure 3). From transcription binding sites predicted within human *SBSN* proximal promoter region by in silico analysis, the SOX2 binding site is currently the only one experimentally validated [12] (Figure 3). Next, a Brother of the Regulator of Imprinted Sites (BORIS)-binding site within the coding region was experimentally verified [13]. Importantly, aberrant changes in expression of *SBSN* isoform are associated with atopic dermatitis (see below [14]), but mechanisms responsible for differential expression of isoforms under physiological and pathological conditions are not known. 

*SBSN* is physiologically expressed in mouse stratified epithelia including the suprabasal epithelial layer of epidermis, tongue, oesophagus, palate, stomach, uterus, thyroid, trachea, lung, vagina, thymus, and urinary tract. In human, *SBSN* expression is associated with epidermis, thymus, uterus, tonsils, vagina, and oesophagus [1,2,15]. Originally, there was no evidence of *SBSN* expression in mouse embryonic and adult brain [1,7,16,17], but recent studies showed, using immunofluorescence, *SBSN* expression in human astrocytes, and its elevation under pathological conditions [18]. Transcriptome analysis of the human brain revealed *SBSN* mRNA levels in basal ganglia, but evaluation of SBSN protein presence in the human brain is needed to confirm this expression site [15]. On the contrary, the mouse brain does not show the presence of *Sbsn* mRNA; neither do several other mouse tissues and organs such as heart, kidney, or smooth muscle [1]. Therefore, between mouse and human, the *SBSN* expression sites seem to be conserved, except for the brain.

Mouse *Sbsn* mRNA was detected on day seven of embryonic development when the expression is likely mediated by extra-embryonic tissue. Hence, the embryonic expression was detected on day 15 of the development at first, which is in coincidence with epidermal stratification. *Sbsn* mRNA levels then peak on day 17 [1], and *SBSN* expression is associated with expression of dermokine-α/β [2], the components of secreted peptides complex. Similarly, *SBSN* mRNA was elevated in skeletal muscle cells of Alaskan sled dogs during an acute response (2 h post-exercise) after a prolonged endurance training together with dermokine and keratin 5 [17]. At the same time point, transcripts of genes involved in inflammation, oxidative stress, intermediary metabolism, immune response, and cellular compromise transcripts, e.g., S100A8, were also upregulated. The role of inflammation, immune, and stress response in *SBSN* expression is supported by the microarray analysis of therapy-resistant cancer cells in vitro, which showed transcript elevation of innate immune response genes and *SBSN* following 5-aza-2′-deoxycytidine (5-AC)-treatment or γ-radiation. Notably, activation of the ERK pathway downstream of IFN signalling emerged as a direct activator of *SBSN* expression [19]. Therefore, epidermal differentiation and response to inflammation provide hints at the understanding of *SBSN* expression inducing processes.

The composition of transcription factors and stimuli responsible for *SBSN* transcription is not specifically defined, but some are suggested. The *Sbsn* transcript elevates in differentiating mouse keratinocytes in vitro, whereas, several genes of the cornified envelope are downregulated upon *SBSN* knockdown [14]. This provides additional support for the coordinated gene expression program during skin differentiation [14]. Targets of the ERK pathway, i.e., components of the AP-1 transcription factor complex c-FOS, FRA-1, FRA-2, c-JUN, JUND, and JUNB, are differentially expressed in keratinocytes during their terminal differentiation in organotypic cultures, and AP-1 proteins are differentially expressed in the human epidermis [20]. This supports the role of ERK in *SBSN* expression; however, the role of ERK in keratinocyte differentiation provides contradictory results [21,22]. Additionally, MAL/SRF signalling also results in *JUNB* elevation, which plays an essential role in epidermal differentiation [21]. Inhibition of BCR-RHOA-MAL/SRF pathway resulted in the reduction of *JUNB* and *SBSN* transcripts, together with disruption of keratinocyte granulation and development of stratum corneum in an organotypic model of the human epidermis [23]. Mouse bearing conditional knockout of *Srf* in basal cells, showed reduced *Sbsn* transcript levels [24], and indeed, the SRF binding site was predicted within the *SBSN* proximal promoter region of both, human and mouse (with three binding sites in mice (−1994/−1978, −1036/−1023, −789/−773 (0.83, 0.8, and 0.8 relative score) and one binding site (−1777/−1760) in human with a 0.82 relative score). Changes in the actin cytoskeleton and MAL/SRF promote physical stimuli-induced keratinocyte differentiation via *JUNB* [21]. Note, in dog muscle, the *JUNB* transcript is elevated 2 h post-exercise, while the *Fos* transcript is downregulated [17]. ERK-mediated regulation of *SBSN* is further supported by increased expression of *Sbsn* in murine endothelial cells following treatment with Egf, but not bFgf [25]. Furthermore, phorbol 12-myristate 13-acetate (PMA)-mediated ERK activation enhanced *SBSN* expression efficiently [1,19], though this effect might be partially mediated by PKC since calcium-induced *SBSN* expression during differentiation of primary epidermal keratinocytes in vitro was suppressed with PKC inhibitor [1,2]. Therefore, multiple pathways activated during keratinocytes differentiation may promote *SBSN* expression likely via, but not only, AP-1-enabled transcription. Indeed, *SBSN* promoter region contains multiple AP-1 binding sites. In total, a JASPAR search predicted 72 AP-1 binding sites, with extensive sequence overlaps, hence lesser number of regions is more likely. Five regions showed >0.9 relative scores (−93/−102, −112/−118, −131/−137, −194/−206, −1147/−1153). 

Lower temperature (33 °C) is frequently used in biotechnology for culture/propagation of Chinese hamster ovary cells. In a recent study, a list of cold-induced genes was established using RNAseq 48 h post change of condition, which included *SBSN* [26]. This was further supported with ectopic expression of luciferase driven by *SBSN* promoter exposed to lower temperatures. *S100A4* was identified to be a cold-induced gene [26], together with *S100A6*, which is supported by a previous study [27]. Furthermore, the JunD binding site was predicted among selected cold-regulated promoters, including *SBSN*, and *JUND* transcript levels were elevated following cold-treatment [26].

In response to hypoxic stress, high altitude acclimatization leads to lower oxygen tension and hypobaric pressure with enhanced hematopoiesis, increased blood volume, and neoangiogenesis to redistribute blood flow to vital organs, including the brain. The latter is mediated by carotid arteries. Upon acclimatization to high-altitude-associated long-term hypoxia, *SBSN* transcript was one of 58 significantly upregulated in carotid arteries in sheep [28]. Interestingly, most altered genes were associated with cell migration, growth and proliferation, and angiogenesis. The authors also noted that some of the regulated genes are also common targets of treatment with lipopolysaccharide (LPS), again supporting the contribution of the innate immune response to *SBSN* expression. Note, the upstream pathway, ERK, and a target of SBSN signalling, AKT, showed increased activation in sheep carotid arteries accompanied long-term hypoxia [28].

The proximal region of *SBSN* gene promoter was originally described as AT-rich, lacking CpG islands and containing a canonical TATA box [1], however, in our analysis presented here (Figure 3) we were not able to identify these regions within 250 bp upstream of +1 site of human *SBSN* promoter region. Conversely, we observed an initiator element at +9/+15 (Figure 3). Multiple NF-kappaB binding sites were predicted within a 2 kbp region upstream of *SBSN* transcription start site in both human and mouse cells [29]. We predicted thirteen NF-kappaB binding sites within this region (Figure 3). The most prominent sites depicted had a relative score > 0.92; −1921/1930, > 0.8; −48/−39). Note, both PMA and LPS are potent inducers of NF-kappaB pathway. Furthermore, several other binding sites for transcription factors, such as SP1, TF2APA, MYC, SMAD2, and FOXO1/FOXO4 were predicted within the *SBSN* proximal promoter [1,30]. Indeed, *SBSN* transcription in confluent human adipocyte tissue-derived stem cells (ASCs) is mediated by FOXO1 [30], since downregulation of FOXO1 with silencing RNA reduced *SBSN* transcript levels. We predicted four FOXO1 binding sites within 2 kbp *SBSN* promoter region (relative score > 0.84; −1616/1609, −1668/−1661, −1692/−1685, −1930/−1923; Figure 3). Interestingly, SBSN promotes aromatase expression, and *SBSN* was shown to be induced with 17beta-estradiol treatment [31]. This is supported with the prediction of estrogen receptor (ER) binding sites within the promoter. We identified thirteen ER binding sites within 2 kbp upstream region of *SBSN* promoter and selected three highest scoring regions to depict (relative score > 0.89; −438/−427, > 0.86; −578/−568, > 0.85; −1902/−1888; Figure 3). Altogether, dozens of transcription factors binding sites are predicted. Needless to mention, confirmation of function of the predicted binding regions require further investigation. 

Multiple studies described aberrant elevation of *SBSN* in human malignancies [19,29,32], but the mechanisms responsible for *SBSN* upregulation under pathological conditions is not understood. As mentioned, *SBSN* expression is limited to specific tissues. Bisulfite sequencing of 11 healthy human lung tissue samples revealed methylation of *SBSN* promoter CpG islands. This indicates that promoter methylation, and likely associated transcriptional repression, are responsible for *SBSN* silencing [33]. Indeed, normal human bronchial epithelial and human small airway epithelial cells treated with demethylating agent 5-AC and histone deacetylase inhibitor trichostatin A (TSA) elevated *SBSN* transcript levels [33]. This is further supported by hypomethylation of *SBSN* promoter in approximately 50% (13/28) of primary non-small cell lung carcinoma (NSCLC) samples observed in the same study. Hypomethylation of the *SBSN* promoter was an effect of a dysregulated proto-oncogenic zinc finger transcription factor, CTCFL/Brother of the Regulator of Imprinted Sites (BORIS) [13]. Similarly, *SBSN* transcription can be induced in vitro in a salivary gland adenoid cystic carcinoma (ACC) cell line with 5-AC and trichostatin A (TSA) resulting in hypomethylation of the CpG island [34]. The *SBSN* transcripts are induced with 5-AC in human cancer cell lines such as DU-145, MCF-7, and HeLa [19]. Notably, SBSN proteins were only detectable in a low-adherent subfraction of therapy-resistant 5-AC-treated cells with stem cell-like properties [19], indicating post-transcriptional regulation of SBSN expression. The *SBSN* gene promoter contains two experimentally confirmed CTCFL/BORIS binding sites in the first exon close to the transcription start site (+202/+374) and in the second intron in front of a CpG island (+2678/+2840). BORIS-mediated induction of *SBSN* is associated with demethylation of a CpG island in *SBSN* second intron and changes in histone marks comprising elevation of the active H3K4me3 and H3K14Ac, and downregulation of the repressive H3K9me3 modifications. Importantly, the elevation of *SBSN* transcript levels mediated by BORIS is dose-dependent. Relatively low BORIS levels were responsible for significantly higher *SBSN* transcript levels compared to high BORIS levels associated with re-methylation of the second intron of *SBSN* and increased nucleosome occupancy of *SBSN* transcription start site. A repressive histone mark H3K9me3 mirrored the *SBSN* second intron methylation pattern. Interestingly, CTCF and BORIS compete for the same binding sites of the SBSN promoter region [13], indicating that the epigenetic and chromatin state play essential roles in *SBSN* expression. Hence, methylation of regulatory sites represents the main feature responsible for the regulation (suppression) of *SBSN* transcription. 

The connection between cell stemness and *SBSN* expression suggested previously [19] is supported by the identification of SOX2, a stem cell factor commonly upregulated in cancer, as a regulator of *SBSN* expression in oesophageal squamous cell carcinoma (ESCC) [12]. SOX2 binding site (−1566/−1559) at the proximal region of human *SBSN* was confirmed by chromatin immunoprecipitation (ChIP), and remains the only determining region of *SBSN* promoter with the validated transcription binding factor. Furthermore, double knockout *Klf2* and *Klf4* mouse cardiac microvascular endothelial cells showed significantly reduced *SBSN* transcript levels compared to control mice [35]. We identified 25 putative KLF4 binding sites and 30 putative KLF2 binding sites in the human *SBSN* promoter region, two most prominent are depicted (KLF4 relative score > 0.9; −250/−239, KLF2 relative score > 0.9; −1993/−1983; Figure 3). These observations strengthen the importance of factors of stemness in *SBSN* expression regulation.

## 5. The Function of SBSN in Context of Physiology and Pathology

As already mentioned, the physiological function of SBSN is currently unknown. *Sbsn* knockout mice do not manifest any abnormal skin phenotype [36], but shRNA-mediated *SBSN* knockdown abrogated the development of *stratum granulosum*, disrupted the formation of keratohyalin granules, and affected the morphology of some keratinocytes in the human living skin equivalent model [37]. The secretory nature of SBSN isoforms is supported by a number of reports [2,7,32,38,39,40], and the SBSN treatment promoted phenotypic changes in vitro [12], indicating the existence of SBSN receptor(s). Studies focusing on the role of *SBSN* in human malignancies provided deeper insight regarding its function. Induction of *SBSN* activated AKT and p38MAPK kinases, and the WNT/β-catenin pathway [12,29]. The activation of WNT/β-catenin pathway was mediated specifically with overexpression of SBSN-2 [29]. And this served as a pro-angiogenic stimulus in mice [29]. Additionally, treatment with recombinant SBSN-2 enhanced sprouting in vitro [12]; thus, SBSN isoform 2 is a proposed oncogene in human malignancies. Therefore, SBSN can act as an oncogenic signalling molecule, but its receptor(s) and downstream signalling pathway(s) are currently unknown. The summary of the described functions of *SBSN* is depicted in Figure 4.

## 6. SBSN in Cancer

The most evidence about *SBSN* is from cancer research. Several studies have suggested the role of SBSN in adaptation to stress conditions [19], activation of pro-surviving signalling pathways [25,29], and angiogenesis [12,29], the features commonly associated with carcinogenesis. The amplification of the 19q13 region, which includes the *SBSN* coding sequence and its proximal promoter, is described in various cancer types, including ovarian, cervical, pancreatic, and breast carcinomas [41,42,43,44].

The first evidence of *SBSN* expression in cancer was a reported increase of *SBSN* mRNA levels linked with the hypomethylation of the *SBSN* promoter in almost half of non-small cell lung carcinoma cases [33]. The ectopic expression of *SBSN* in lung squamous cell lines resulted in increased anchorage-dependent growth in soft agar assay. The expression of *SBSN* in lung cancer correlates with BORIS [13,33]. BORIS is a transcription factor specific for the male germ line and is implicated in the activation of cancer-testis antigen (CTAs) genes [45,46,47]. BORIS is aberrantly expressed in several types of human cancers such as lung, head and neck, breast, and bladder carcinomas [48]. Additionally, *SBSN* CpG island in the second intron is significantly hypomethylated in primary salivary gland adenoid cystic carcinoma (ACC) compared to the normal salivary gland tissue. Knockdown of *SBSN* in a salivary gland ACC cell line suppressed anchorage-dependent growth in the soft agar and invasiveness in Matrigel invasion assay. Nevertheless, no correlation of SBSN with gender, smoking history, tumour or nodal stage, or metastatic disease of ACC was found. Also, no correlation with patient survival or time to disease relapse were found [34]. 

The oncogenic role of *SBSN* was originally proposed in a study of oesophageal squamous cell carcinoma (ESCC) [29]. In comparison to normal oesophageal epithelial cells, 11 ESCC cell lines showed elevated *SBSN* mRNA and the 25 kDa protein isoform. Importantly, ESCC tissue samples also showed elevated *SBSN* mRNA levels compared to unaffected tissue. Immunohistochemical analysis of 170 clinical ESCC specimens revealed a positive correlation between *SBSN* expression and tumour size, tumour clinical stage, and patient vital status. Patients with higher *SBSN* expression showed shorter overall survival [48]. A subsequent study suggested SBSN as a potential biomarker of ESCC [49]. The overexpression of *SBSN* isoform 2 promoted proliferation and anchorage-independent growth of ESCC cell lines in vitro (and normal human oesophageal epithelial cells as well) and tumour growth in vivo, whereas *SBSN* knockdown showed the opposite effect. SBSN-2 in ESCC cell lines increased the activity of WNT/β-catenin signalling pathway, which resulted in TCF/LEF transcriptional activity, nuclear translocation and reduced phosphorylation of WNT/β-catenin, and increased transcript levels of WNT/ β-catenin signalling-regulated genes such as *AXIN2, MYC*, *CCND1, FRA1, MMP7* and *JUN*. The effect of *SBSN* on WNT signalling was mediated by phosphorylation of GSK3 β. WNT/ β-catenin signalling pathway regulates several cancer-associated processes such as cell growth and cell death, migration, invasiveness, stemness, and differentiation. Again, *SBSN* expressing ESCC cell lines formed in vivo tumours with higher microvascular density indicating the role of *SBSN* in angiogenesis [29]. In line with these observations, the treatment with recombinant SBSN-2 induced migration and sprouting of HUVEC [12]; however, this was accompanied by activation of AKT and p38MAPK. The phenotypic changes in the cells line were predominantly mediated by activated AKT since p38MAPK inhibition did not abrogate SBSN-2-promoted effects, whereas AKT inhibition did. Therefore, SOX2-regulated *SBSN* was suggested to mediate the angiogenic potential of early-stage ESCC via AKT signalling [12]. 

Tumour growth depends on *de novo* angiogenesis in tumour tissue. In accordance with the effects of *SBSN* in angiogenic processes [12,29], human tumour endothelial cells (TEC) isolated from renal cells and colon carcinomas expressed higher levels of *SBSN* compared to non-tumour (NEC) endothelial cells. *SBSN* was upregulated in mouse TEC isolated from human tumour xenografts, and its knockdown significantly suppressed VEGF-A-mediated migration and tube formation of mouse TEC compared to mouse NEC. Moreover, *Sbsn* knockdown reduced the phosphorylation status of Akt, but not Erk kinase, in mTEC indicates that Sbsn may take part in TEC migration, tube formation and pro-angiogenic activity via Akt activation [25]. The link of *SBSN* expression to vein cell function is further supported by an observation that knockdown of *SBSN* affected the capability of vein wall-derived cells to contract collagen gel in vitro [50].

In the case of malignant brain tumours, the higher *SBSN* transcript levels correlated with significantly lower survival of glioblastoma multiforme (GBM) patients. *SBSN* transcript levels are elevated in a GBM subtype with mesenchymal signature, the phenotype associated with a strong immunosuppressive milieu, therapy resistance, and the elevation of angiogenic markers [51,52]. Utilizing a mass spectrometric Stable isotope labelling with amino acids in cell culture (SILAC) approach, the secretome analysis of four glioblastoma cell lines (LN18, T98, U118 and U87; [38]) revealed SBSN peptides in the secretome of U87 together with other invasive factors. U87 was the most invasive cell line in a Matrigel invasion assay, but the role of *SBSN* was not determined. In another glioblastoma cell line U373, the siRNA-mediated knockdown of *SBSN* expression its participation in radioresistance via an unknown mechanism [19]. 

The role of *SBSN* in therapy resistance is supported by other findings. The expression of *SBSN* following radio- or chemo-therapy regimes in stem-like cells with a low-adhesive phenotype was mediated by ERK1/2 activity in breast, prostate and cervical cancer cell lines [19]. Knockdown of *SBSN* relieved ERK1/2 activity-dependent anoikis resistance of low-adherent cells. Similarities in a gene expression pattern in differentiating mouse skin [16] and irradiated low-adherent cells [19] can be observed. For instance, transcription factors Klf4, Klf6, Cebp α/β, Fos, Jun, JunB and Cdk inhibitor p21waf1 were upregulated during embryonic development of epidermis at days 14.5–15.5 and were detected in the transcriptome of low-adherent cancer cells as well. This indicates that the keratinocyte differentiation programme can take part in the development of low-adherent anoikis-resistant state of human cancer cells induced with genotoxic stress. The loss of adhesion, and, in some cases, nuclear degradation underscores the presence of keratinocyte differentiation-like regulatory network following therapy-induced phenotype. 

*SBSN* was identified among 16 genes negatively associated with lymph node metastases in head and neck cancer (HNC) patients [53]. In this study, cornifelin (*CNFN*) represented a hub gene of the *SBSN*-containing gene cluster. This cluster was associated with gene set enriched for the p53 pathway and estrogen pathway genes. This suggests that the *CNFN* and *SBSN*-16-genes containing set could be functionally connected [53]. Previously, the estrogen biosynthesis and *SBSN* expression were linked via elevation of aromatase transcript levels in confluent ASCs [30] and induction of *SBSN* expression following 17beta-estradiol treatment of immortalized vaginal epithelial cells [31]. Importantly, SBSN in ASC-conditioned medium promoted migration of ER-negative breast carcinoma cell line MDA-MB-231 in an estrogen-independent manner [30]. Note, aberrantly elevated SBSN is observed in the bone marrow, and the peripheral blood of myelodysplastic syndromes (MDS) patients [32], hence systemic presence of SBSN accompany this disorder. 

MDS represent a heterogeneous group of pre-leukemic diseases. MDS are characterized by ineffective haematopoiesis affecting the myeloid lineage predominantly [54]. Two studies showed the presence of SBSN in the blood of MDS patients. Firstly, SBSN was identified as an interactor of LRG1 derived from peripheral blood (PB) serum of MDS patients [55]. Note, LRG1, a granulocytes-progenitor receptor [56], was confirmed in neovascularization processes of retinal disease [57]. Secondly, *SBSN* expression was shown to be aberrantly elevated by myeloid compartment, predominantly by myeloid-derived suppressor cells (MDSCs) and early-stage MDSCs (eMDSCSs), in bone marrow (BM) of MDS patients [32]. Importantly, a subgroup of MDS patients with the highest bone marrow (BM) SBSN levels corresponded to a poor prognosis, high-risk group. *SBSN* expression was therapy-independent, and hence an intrinsic feature of the disease. Importantly, SBSN levels in peripheral blood reflected BM levels of SBSN [32]. Together with the observation of the presence of SBSN in the pleural fluid of patients suffering from lung adenocarcinoma [39], MDS represent another human malignancy in which SBSN is proposed as a potential biomarker [32]. Furthermore, SBSN aberrantly *O*-glycosylated at threonine 221 was observed specifically in peripheral blood (PB) plasma of gastric cancer patients [58] and reduced PB SBSN plasma levels in patients with atopic dermatitis [37] support that dysregulated SBSN plasma levels reflect disease conditions. Furthermore, BM SBSN showed a negative correlation with BM T cell counts and CCL2 levels in MDS, implicating anti-inflammatory conditions which might impede potentially the immunological response [32]. Importantly, *SBSN*-expressing BM MDSCs play a crucial role in the pathogenesis of MDS [59]; hence novel therapeutic strategies strive to target this cellular compartment [54]. Among multiple other factors, MDSCs-secreted alarmins S100A8/9 are mediators of the inflammatory response and subsequent pathogenicity mediated by MDSCs [60], this further supports the role of *SBSN* as a component of the innate immune response. 

There is also limited evidence for a possible association of *SBSN* with ovarian carcinoma. Ovarian carcinoma derived SK-OV-3 cell line expresses *SBSN* in vitro and marked elevation of *SBSN* transcripts was found in patients’ biopsies of ovarian tumours [19]. These results indicate a potential function of *SBSN* in ovarian carcinoma, but the association of SBSN with the prognosis of ovarian cancer requires further investigation. 

The relationship of *SBSN* expression with metastatic stages of malignant diseases is supported by a study employing mouse 4T1 cancer model. The blood-circulating 4T1 cells and 4T1 cells isolated from liver metastases showed higher *Sbsn* transcript levels compared to the primary tumour or parental 4T1 cells [19]. Furthermore, the link of *Sbsn* to metastatic disease is supported by another study [61]. To elucidate the role of T cell activity in the development of HNSCC, the microarray analysis of primary and metastatic sites in DMBA-induced HNSCC of nude and C57Bl6J mice were compared. The primary SCC in nude mice showed reduced transcript levels of terminal differentiation genes, such as *Tgm3*, *desmoglein 1beta*, and *corneodesmosin* specifically in nude mouse. Interestingly, histopathology revealed a difference in cervical lymph node metastatic tumours of wild-type and nude mice, showing reduced keratinization potential of these tumour sites developed in nude mice. This was accompanied with markedly reduced *Sbsn* transcript levels in metastatic SCC in nude mice. Comparison of primary and metastatic tumours cells developed in nude mice showed even stronger reduction in *Sbsn* expression [61]. Since T cell maturation-lacking nude mice showed markedly reduced expression of *Sbsn* in DMBA-induced HNSCC primary tumour cells and metastatic cells compared to C57Bl6J mice [61], we can speculate that *Sbsn* expression is a response of cancerous cells to anti-tumour T cell activity. 

## 7. SBSN in Other Human Pathologies

With the onset of mass spectrometry (MS)-based proteomics, large datasets opened new avenues in understanding human diseases. Besides cancer, a number of studies showed the presence of SBSN peptides also under other pathological conditions, but only limited peptide counts and coverage of SBSN protein have been achieved [8,40,62]. The possibility of multiple post-translational modifications, and a therefore limited chance of identification, cannot be disqualified. Additionally, the combined analysis of SBSN mRNA and protein levels should be approached in subsequent studies. 

To investigate the pathology of Fuchs endothelial corneal dystrophy (FECD), label-free quantitative tandem mass spectrometry proteome analysis was performed revealing SBSN peptides specifically in the aqueous humor of FECD patients [63]. FECD is an eye disorder causing corneal oedema and clouding, which results in vision impairment. FECD aetiology is unknown; however, the progressive loss of corneal endothelium is observed at the onset of the disease. Endothelial cell apoptosis likely promotes the synthesis of extracellular matrix (ECM) proteins responsible for tissue deformation and functional disruption [64]. Next to presence of SBSN peptides, downregulation of complement C3 protein and FAM3C (also downregulated in sporadic Alzheimer’s disease [65]) were identified as significant hits. Interestingly, spaceflight-exposed mouse model showed elevated *Sbsn* transcript levels in the retinal tissue [66], but this was not supported by later proteomic studies [67,68]. Similarly, the transcriptome analysis of cornea affected with fungal keratitis, a type of microbial infection which may eventually lead to loss of vision, revealed marked upregulation of anti-microbial, innate immune defence, wound healing, and cornified envelope genes. The latter included *SPRR2A, SPRR2D, SPRR2F, SPRR3,* and *SBSN* [69]. In line with described changes, several MMPs and pro-inflammatory cytokines were also elevated. 

A follicular fluid isobaric tags for relative and absolute quantitation (iTRAQ) proteome study of polycystic ovary syndrome (PCOS) patients revealed downregulation of complement components and dysregulation of proteins functionally associated with angiogenic processes, for instance, PLG, AGT, LYVE1 (pro-angiogenic), and SERPINA1 (an inhibitor of PLG; anti-angiogenic). Importantly, the follicular fluid of PCOS patients was enriched in SBSN peptides [40]. Note, chronic inflammation, endothelial dysfunction, and elevated risk of cardiovascular disease, are commonly associated with PCOS [70,71,72].

In a rat model of pre-motor Parkinson’s disease (PD), the blood plasma proteome analysis of citrullinated proteins revealed a Sbsn peptide among PD-specific hits [8]. Citrullination is physiologically mediated by peptidylarginine deiminases (PAD). PAD gene expression has been shown in several organs but not in the thymus; therefore, citrullinated peptides are not subjected to central immune tolerance [73,74]. However, this may lead to neoantigen production and subsequent autoimmune recognition, the cause of potential autoreactivity. Pathological peptide citrullination is commonly associated with chronic inflammatory diseases and therefore considered as a response to chronic inflammation [74]. Citrullinated proteins recognized in other neurodegenerative diseases, including Huntington’s and Alzheimer’s diseases, suggest the role of aberrant peptide citrullination in the onset of these pathologies [73].

Another source of pathological citrullinated peptides is *Porphyromonas gingivalis*-expressed peptidylarginine deiminase (PPAD). Besides PPAD, *P. gingivalis* secretes a group of arginine-/lysine-targeting proteases, gingipains. Cleavage of host proteins with subsequent PPAD-mediated citrullination may lead to autoimmune response via antibodies recognizing the modified peptides. *P. gingivalis* is commonly associated with chronic periodontitis and oral squamous cell carcinoma [75]. Transient bacteremia may lead to transition of *P. gingivalis* to arteries [76], and arterial colonization with *P. gingivalis* is associated with cardiovascular diseases [77]. Additionally, *P. gingivalis* proteins are present in the majority of Alzheimer’s disease patients’ brains and correlate with the disease progression [78], and gingipains triggered Alzheimer-like disease phenotype in induced pluripotent stem cells-derived neurons [79]. Recently, Liu et al. treated oral squamous cancer cells with inactive *P. gingivalis* showing elevated *SBSN* mRNA levels 16 h post treatment. Additionally, tumour-associated macrophages showed pro-tumorigenic M2-transition following the bacterial treatment [75]. Altogether, these observations indicate that chronic inflammation and *P. gingivalis* infection could promote *SBSN* expression. Certainly, elucidation of the role of *SBSN* or presence of antibodies recognizing SBSN represent the next steps in further investigations. 

The analysis of MHC II immunopeptidome revealed an abundance of SBSN peptides in axillary, brachial, inguinal, and skin-draining lymph nodes of healthy mice [62], indicating the importance of peripheral lymphoid organs in developing immune tolerance towards SBSN peptides, or posttranslationally modified SBSN peptides. 

Indeed, in a recent search for neuropsychiatric systemic lupus erythematosus (NPSLE)-specific autoantibodies in cerebrospinal fluid (CSF) that could potentially be used as diagnostic markers of the disease, SBSN peptide-recognizing immune complexes were identified [18]. This indicates the presence of anti-SBSN autoantibodies specifically in a subgroup (nine out of 26 patients, only females) of NPSLE patients [18]. No SBSN immune complexes were found in the peripheral blood plasma of any NPSLE patients; therefore, the production of anti-SBSN autoantibodies by brain parenchyma-infiltrating lymphocytes intrathecally was suggested. However, the observation of SBSN immune complexes in CSF does not exclude the presence of anti-SBSN autoantibodies in the patients’ serum without the immune complexes formation. Additionally, authors showed higher *SBSN* expression in hippocampal astrocytes of NPSLE patients compared to healthy individuals where SBSN was also detected. Once again, these observations show aberrant *SBSN* expression under pathological conditions. Besides others, CSF of NPSLE patients is enriched in inflammatory cytokines [80], including IFN-gamma [81], a recently identified inducer of *SBSN* expression in cancer cells [19]. Interestingly, LPS-stimulated human normal astrocytes treated with commercially available anti-SBSN antibody showed elevated transcript levels of a few genes associated with senescence and autophagy, and TGF-beta signalling pathways [18]. 

## 8. SBSN, Bacterial Infection, Immune Response, and Obesity

As described above, *SBSN* transcript elevation follows *P. gingivalis* stimulation of oral cancer cells [75]. *SBSN* implication in bacterial response seems to be more profound, but yet not clear. 

*Trypanosoma cruzi* represents a pathogen with an ability of congenital transmission. To elucidate the response of human prenatal tissue to *T. cruzi* infection, human placental extract (HPE) was infected with trypomastigotes of *T. cruzi*. Two hours post-treatment transcripts of pro-inflammatory cytokines, TLRs, and NLRs were elevated, but, 24 h post-infection, the *SBSN* transcript was among top upregulated genes [82]. Furthermore, mass spectrometry analysis of extracellular vesicles derived from human macrophages revealed SBSN peptides following LPS treatment (TLR4-stimuli; [83]). Additionally, human peripheral blood mononuclear cells (PBMC) treated with tRNA or 16 kDa lipoprotein (TLR1/2-stimulus) of *Mycobacterium tuberculosis*, or ssRNA (HIV-I derived sequence, TLR8-stimulus) significantly elevated *SBSN* mRNA levels after 6 h [84]. These results suggest *SBSN* expression can be a direct response following TLR receptors stimuli, and a downstream target of NF-kappaB as already proposed [29]; therefore *SBSN* is component of innate immune response and/or early anti-microbial defence.

Transcriptome analysis of vaginal epithelial cells from post-menopausal women with vaginal dryness, symptom commonly associated with vaginal microbial infection and epithelial irritation, showed downregulation of *SBSN* and *TGM3* [85]. Gene ontology analysis revealed downregulation of cornified envelope, keratinocyte differentiation, immune response, and innate immune response genes in symptomatic women. In opposite, uropathogenic *Escherichia coli* (UPEC) colonizing the lumen and the epithelial cells of the bladder and responsible for chronic bacterial infection of the urinary tract, initiated innate immune defence and epithelial cell detachment [86,87]. *Sbsn, Ivl, Lcn2,* and complement components B and C3 were markedly elevated by murine urothelial cells in close proximity of intracellular UPEC colonies [88]. These results are supported by a subsequent in vitro study [31], in which the treatment of immortalised vaginal epithelial cells with 17beta-estradiol, flagellin of *E. coli*, or both, increased transcript levels of innate immune response, keratinization, and cell differentiation genes. *LCN2*, *S100A8/9* (essential mediators of MDSC response in MDS [59]), *SPRR2A* and *SPRR2F* (essential in cornification processes [89]) were elevated with each treatment. *SBSN* together with *CNFN*, *LCE3D*, *SPLI* (cornification processes)*,* and *F3* were significantly elevated following 17beta-estradiol or 17beta-estradiol plus flagellin treatments. Besides others, 17β-estradiol elevated *IL-17A* transcript in the vaginal biopsies of treated post-menopausal women, implicating the stimulation of anti-microbial defence. Additionally, transcriptome analysis of mammary gland parenchyma of intact and ovariectomized prepubertal heifers revealed *SBSN* among estrogen-regulated genes [90]. 

Obesity is a comorbidity associated with multiple human diseases, including cancer. Chronic systemic inflammation underlies obesity-related symptoms [91]. In an attempt to elucidate the inflammatory role of human adipose tissue-derived stem cells (ASCs) under cell confluency in vitro, *SBSN* was found to be a FOXO1-regulated gene with a capacity to promote aromatase (*CYP19A1* gene) expression. Similarly, FOXO1 depletion resulted in downregulation of *CYP19A1* transcripts under confluent conditions. The ability of ASCs to express SBSN and aromatase was donor-dependent, hence likely dependent upon epigenetic state and priming of the cells [30]. The FOXO1-SBSN-aromatase pathway thus possibly results in elevation of estrogen levels, and as described above, 17beta-estradiol-induced *SBSN* transcription in vaginal epithelial cells. This implicates a positive feedback loop which could result in dysregulation of estrogen levels and SBSN-mediated response. 

Despite its high prevalence, especially in children, atopic dermatitis (AD) remains a poorly understood and inefficiently treated immune system-driven cutaneous disease. Dysregulated immune cell-mediated cytokine signalling affecting the differentiation of epidermal cells as one of the features of AD is detrimental and predominantly driven by Th2-expressed *IL-4* and *IL-13* genes [92,93]. In the murine model, IL-4 was shown to downregulate the expression of epidermal barrier genes, *Flg* and *Ivl* [94]. Similarly, in vitro studies using NHEK revealed slight downregulation of *SBSN* mRNA following IL-4 treatments [14], but, human living skin equivalents did not reproduce these results following IL-4 treatment [37]. Healthy skin possesses mostly transcripts of *SBSN* isoform one, whereas levels of transcripts of isoform two and three are lower [14]. Interestingly, transcripts of all *SBSN* isoforms were markedly elevated in non-lesional skin samples of AD patients, but transcripts of all isoforms were reduced in skin lesion samples of AD patients [14]. The proteome analysis of skin tissue revealed a significant reduction of SBSN in AD patients as compared to healthy tissue. The lesional skin of AD patients showed varying SBSN staining intensities, including cases with low SBSN level staining as well as cases with SBSN intensities comparable to healthy tissue. Furthermore, AD and psoriasis vulgaris patients were characterized with reduced SBSN blood plasma levels, and patients with intrinsic AD type (non-systemic response), showed even reduced SBSN serum levels compared to extrinsic AD type (systemic, high serum IgE). shRNA-mediated *SBSN* knockdown induced apoptosis of fraction of keratinocytes in human living skin equivalent model, and IL-4/IL-13-treatments enhanced the observed apoptotic features, indicating SBSN-mediated resistance to IL-4/IL-13-induced apoptosis in human keratinocytes [37].

## 9. Conclusions

*SBSN* remains a gene of unknown molecular function. SBSN proteins represent secretory molecules with the ability to activate the signalling cascades, but its receptor(s) are not yet identified. Based on current evidence, *SBSN* is associated with multiple human diseases, including cancer and autoimmune disorders. Various cellular compartments express *SBSN* in diseases, hence uncovering stimuli and conditions responsible for *SBSN* expression in different cell types would provide valuable knowledge. Several *SBSN* expression-inducing stimuli are already confirmed (such as PMA, EGF, γ-radiation, and IFN-gamma). Also, other stimuli lead to *SBSN* transcript elevation (endotoxins and inflammatory signals), but their *SBSN*-inducing mechanisms remain to be elucidated. *SBSN* expression is likely post-transcriptionally regulated, and SBSN undergoes multiple post-translation modifications, which suggests a complex nature of *SBSN* biology. Importantly, dysregulated *SBSN* expression and SBSN plasma level alterations mark malignant conditions, potentially promoting systemic effect and patients response to the therapy [30,32,37]. Therefore, SBSN possesses the potential to serve as a biomarker of various human diseases [32,37,39] which has already been proposed [32,39]. 

Several cellular sources of *SBSN* expression were documented. Cancer cells expressing *SBSN* display stem markers [12,19] and indeed, SOX2 is currently the only identified transcription factor known to promote *SBSN* expression specifically in ESCC [12]. Tumour endothelial cells express *SBSN* in colon cancer and the effect of SBSN on endothelium is indicated in several reports [12,25], suggesting *SBSN* might play a role in the response of endothelial cells to aberrant conditions. Chronic inflammatory signalling accompanies the onset of tumorigenesis, promotes stemness [95], and the development of MDSCs [96]. MDSCs are known to mitigate immune response efficiently. Interestingly, MDSC-expressed *SBSN* was negatively correlated with BM T cell abundance in MDS [32]; hence SBSN could contribute to immune evasion of cancerous cells. These observations indicate that tumour microenvironment is causal for the induction of *SBSN* expression in different cell types of the malignant landscape.

The role of *SBSN* in immune regulation is strengthened with recent observation in *Sbsn*-null mice [36]. This animal model showed no aberration in skin development. However, following the low-dose nickel challenge, the mice did not develop T regulatory cells (Tregs) in the spleen, suggesting that *SBSN* is crucial in Treg development induced with these stimuli [36]. Other possibilities, such as induction of Treg development in general or induction of anergy in the tumour microenvironment specifically need to be evaluated in future studies. 

In search for *SBSN*-inducing stimuli, we compiled various observations, indicating that innate immune response is a likely candidate [84,85]. *SBSN* transcripts elevated following treatments with bacterial components or lysates in different animal models [84,85]. Other genes of the innate immune response are upregulated together with *SBSN* transcripts, for example, *S100A* family representatives. Innate immune response accompanies early stages of tumour development associated with microenvironment remodelling and a dysregulated immune response [97]. Note, MDSCs-driven aberrant innate immune response in MDS BM contributes to the disease pathogenesis [59].

The fundamentals of *SBSN* are not yet understood; therefore, *SBSN* represents an interesting target for further studies in both physiology and pathology. We predict that *SBSN* will emerge in other diseases as a candidate diagnostic and prognostic biomarker. Understanding its function is of great importance since *SBSN* likely plays a role in disease development. 

## 10. Future Perspectives

There is a plethora of questions regarding *SBSN* biology that remains to be answered:
(1)Is SBSN a ligand and an inducer of signalling pathway?

Although the molecular function of *SBSN* has not been identified, current findings of SBSN secreted nature and the effects in experiments in vitro using recombinant SBSN indicate that SBSN can function as a ligand via an unknown receptor. The interacting partners of SBSN were not studied specifically. LRG1 remains to be the only shown interactor in peripheral blood of MDS patients [55]. Furthermore, the mechanism of activation of signalling AKT, p38MAPK, and/or WNT/β-catenin pathways or other pathways should be explored.
(2)What stimulates aberrant expression of *SBSN* under malignant conditions?

Several signals that could promote aberrant *SBSN* expression associated with human malignancies were reported. Some of these are already documented with a degree of certainty, such as EGF [25], IFN-gamma [19]; however, the exact molecular mechanisms remain to be investigated. Stemness seems to be an important factor in *SBSN* regulation [12,19]. Additionally, cancer therapy itself (e.g., γ-radiation or 5-AC) may promote *SBSN* expression [19]. Multiple reports showing *SBSN* transcript elevation should be broadened by uncovering the *SBSN* molecular role, for instance, in innate immune response [31,84,88], estrogen [30,90], and temperature alterations [26]. Importantly, the mechanisms of SBSN post-transcriptional regulation, as observed in therapy-resistant cancer cells [19], need to be elucidated.
(3)What is the functional role of individual SBSN isoforms?

Different protein isoforms may possess diverse and contradictory functions, and this cannot be excluded in case of *SBSN.* The oncogenic potential of *SBSN* is linked to human *SBSN-2* [12,29] and should not be extrapolated to other isoforms unless proved experimentally. Indeed, SBSN-1 was shown to be expressed in therapy-resistant non-cycling tumour cells [19]; therefore, it could be speculated that human SBSN-1 and -2 promote contradictory effects. Post-translational modification of the isoforms can also alter SBSN-mediated effect. Strikingly, rodents lost the equivalent of human SBSN isoform 2, therefore elucidating SBSN-2 function utilizing rodent models is restricted to ectopic expression of human SBSN-2. 

## Figures and Tables

**Figure 1 genes-12-00108-f001:**
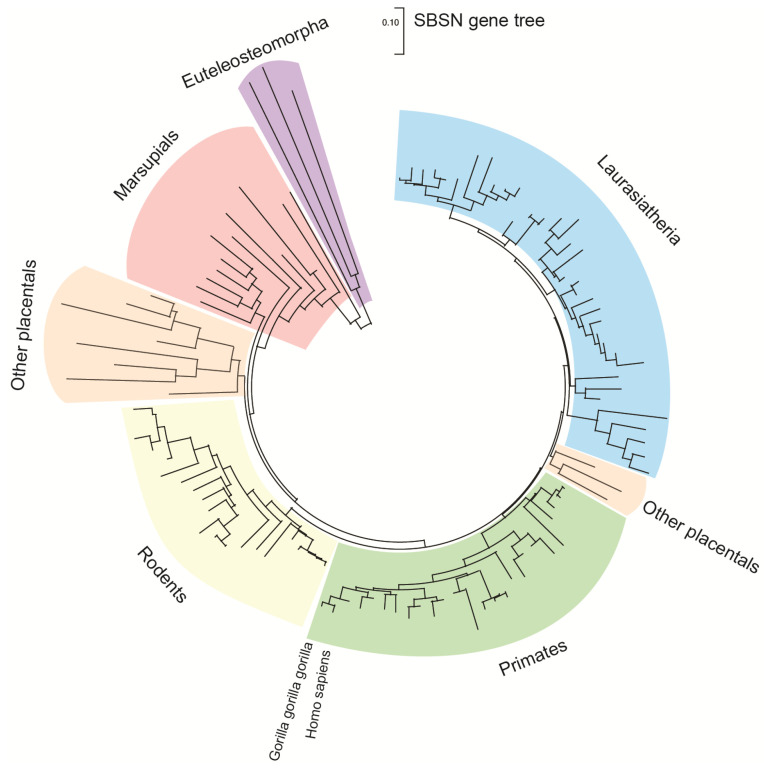
The gene tree of suprabasin *SBSN*. The gene tree of *SBSN* generated with MEGA X software [4] utilizing Enesembl (version 101) available data [3]. With the total sum of branch lengths (SBL) being >9.477. Distance scale = 0.10.

**Figure 2 genes-12-00108-f002:**
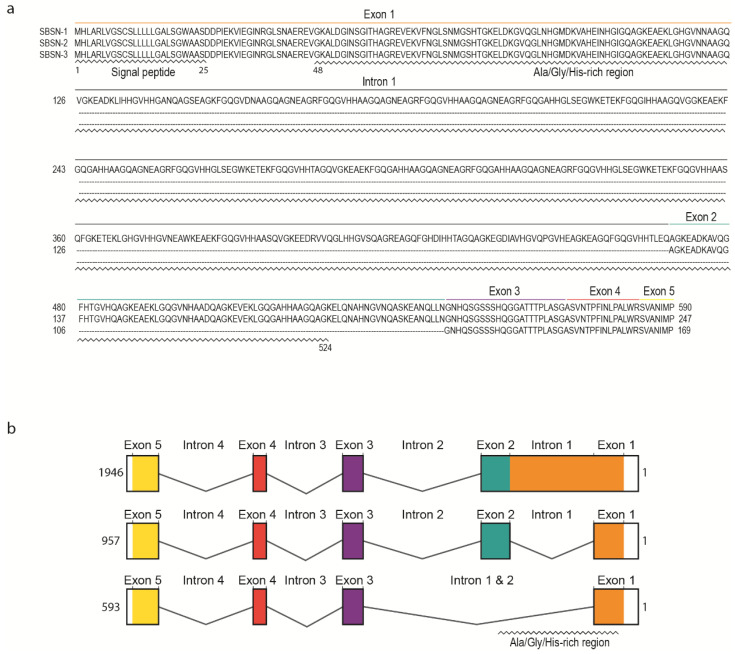
The sequences of SBSN isoforms. (**a**) The amino acid sequences of human SBSN isoforms. (**b**) The gene organization of human SBSN isoforms.

**Figure 3 genes-12-00108-f003:**
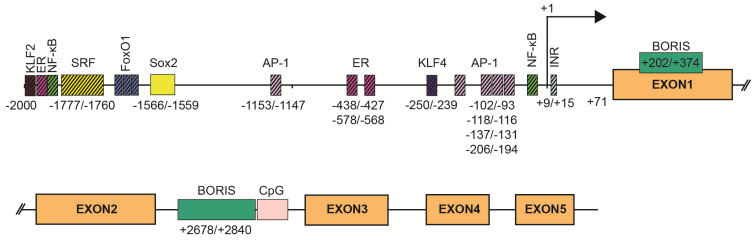
*SBSN* promoter. Two kilobase pairs upstream of the promoter and downstream sequence of human *SBSN* with predicted transcription factor binding sites. SOX2 binding site is currently the only validated site [12] together with Brother of the Regulator of Imprinted Sites (BORIS) binding sites [13].

**Figure 4 genes-12-00108-f004:**
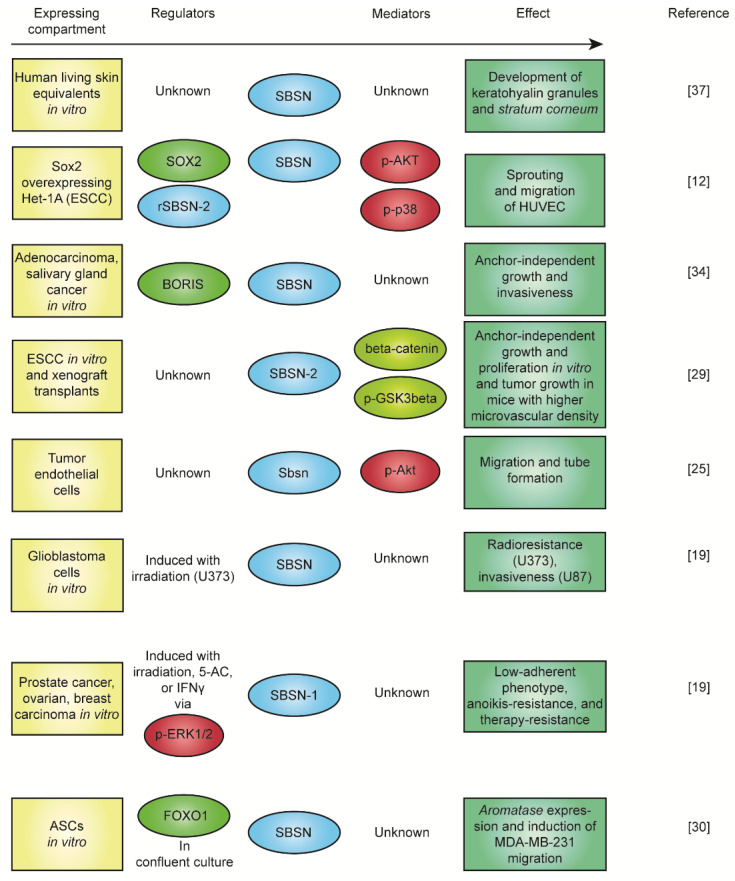
SBSN and observed effects in various models. SBSN and observed effects based upon results from compiled literature. ESCC: oesophageal squamous cell carcinoma; HUVEC: human umbilical vein endothelial cells; ASCs: adipocyte tissue-derived stem cells.

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
