# Peer review of "Suprabasin—A Review"

_genes, 2021, doi:10.3390/genes12010108_

Round 1
Reviewer 1 Report
The Authors presented a extensive review of Suprabasin. As the gene was associated with multiple pathologies and diseases, its precise function remains elusive. The review has defined the current knowledge in the field of SBSN in all aspect presented in literature so far. All data were supported by highly selected references.
I have no major remarks, but the organization of paper is confusing. The 1. Introduction section is too long. The subsections should be presented as single sections. This reorganization will make the manuscript more transparent.
Minor remarks:
Line 37: therapy, , : unnecessary space
Line 47: desmokin- / : no characters
Line 59: there is no reference to database which is recommended in “Genes” Instructions for Authors (Title of Site. Available online: URL (accessed on Day Month Year).
Line 75: there is no reference for SBSN-2
Line 80: redundant paragraph
Line 93: redundant paragraph
Line 103: histidine- ….: unnecessary space; GxHH- please explain
Line 106: redundant paragraph
Line 121: reference absence
Line 126: there is no reference to UniProtKB database
Line 129, 144, 170, 171: redundant paragraph
Line 189: dermokine-…: no characters
Line 235: CHO cells: please explain
Line 236: redundant paragraph
Line 237: supported
Line 257: please explain, what is the reason of differences in SBSN gene promoter descriptions
Line 258, 262: no characters
Line 314: transcription binging factor
Line 321: 1.4. Section: change the font size
Line 336: The description of Figure 4 should be shortened (without repeats).
Line 360: ESCC- please explain
Line 361: redundant paragraph
Line 371, 373: Wnt/…, -catenin- no characters
Line 435, 436, 437, 441, 570, 592: redundant paragraph
Line 667: various
Line 695: …-radiation: no characters
Abbreviations: Wnt Wnt signaling pathway?
Author Response
At first, we are grateful to the reviewers for their time and constructive comments on our manuscript. We have implemented their comments and suggestions in the revised version of the manuscript. New additions to the initial version of the manuscript are highlighted in yellow in the revised version of the manuscript. Below, we also provide a point-by-point response (in blue) on how we have addressed each of the reviewer’s comments and questions.
We hope the reviewers will find our respond satisfactory and the revised manuscript suitable for publication in Genes.
Reviewer 1:
We have removed all redundant paragraphs and spacing as suggested by the reviewer.
We have substituted the missing characters as pointed out by the reviewer in line 47 (dermokine-alpha/beta at page 2 and page 6), page 7 (NF-kappaB), and on the page 10 WNT/beta-catenin and GSK3beta. We have decided to use the words instead of the symbols to overcome the changes mediated by different font types.
The 1. Introduction section is too long. The subsections should be presented as single sections. This reorganization will make the manuscript more transparent.
Re: We thank the reviewer for an excellent idea. We have decided to modify the introduction according to the reviewer’s suggestion. The subsections of the introduction are currently presented as individual sections, thus reducing the length of the introduction which is currently in a single paragraph. Furthermore, we have shortened the length of the introduction. The current form of the introduction claims:
“Since its original description in human and mouse keratinocyte differentiation [1], suprabasin (SBSN) was associated with multiple diseases, including cancer. SBSN isoforms are putative signalling molecules inducing cellular signaling (AKT, WNT/beta-catenin, and/or p38MAPK signaling) and various cellular processes, such as migration, proliferation, neovascularization, therapy-, apoptosis- and immune-resistance. Therefore, SBSN is considered as an oncogene, and is a proposed biomarker in a couple of diseases, lung carcinoma and myelodysplastic syndromes (MDS). The apparent need for a deeper understating of the nature and the function of SBSN prompted us to compile all current knowledge of SBSN together with suggestions for future research work.”
Line 59: there is no reference to database which is recommended in “Genes” Instructions for Authors (Title of Site. Available online: URL (accessed on Day Month Year). And Line 126: there is no reference to UniProtKB database.
Re: We thank the reviewer for pointing out the missing information. The reference to Ensembl database has been updated according to “Genes” instruction recommended by the reviewer. Currently, the reference is formed as: Ensembl Genome Browser version 101, Available online: https://www.ensembl.org/index.html, accessed on 21st August 2020.
The reference to UniProtKB database has been updated according to “Genes” instruction as recommended by the reviewer. Currently, the reference is formed as The UniProtKB release 2020_4, Available online: https://www.uniprot.org/, accessed 21st August 2020. The reviewer suggests the correction on the page 5, however, the UniProtKB is mentioned on the page 3 already. Hence we added the reference on the page 3, where the UniProtKB is firstly mentioned. A reference for UniprotKB has been added as well.
Line 75: there is no reference for SBSN-2
Re: We have added a new citation originally identifying the cDNA of SBSN isoform 2 (Clark, 2003).
Line 103: histidine- ….: unnecessary space; GxHH- please explain
Re: We have utilized GxHH as a short version for newly added Glycine-x-Histidine-Histidine, while stating, that x stands for any classical amino acid.
Line 121: reference absence
Re: A reference has been added in the line 121 ((Park et al., 2002)
Line 235: CHO cells: please explain.
Re: The Chinese hamster ovary has been substituted for CHO abbreviations, and this abbreviation has been removed from the list of abbreviations, since it is not used anywhere else in the text.
Line 257: please explain, what is the reason of differences in SBSN gene promoter descriptions
Re: The authors in Park et al. 2001 described SBSN promoter as AT-rich, without describing the method or analysis applied. We manually searched for AT-rich regions within proximal upstream regions of SBSN promoter, without success. We were not able to identity AT-rich regions commonly associated with promoter regions. We only identified the initiator element which we described and depicted in figure 3.
Line 336: The description of Figure 4 should be shortened (without repeats).
Re: The description of Figure 4 has probably been mistaken for the paragraph in the section 5. We changed the font size of the paragraph in order to make it distinguishable from the description of the figure 4.
Line 360: ESCC- please explain
Re: ESCC is explained on the page 8. But we decided to repeat the description of the abbreviation on the page 10 as the reviewer suggests.
Abbreviations: Wnt Wnt signaling pathway?
Re: Wnt – Wingless-type abbreviation is at the very last spot of the list of abbreviations.

Reviewer 2 Report
Pribyl et al. provide a comprehensive review on the gene SBSN, highlighting current knowledge on isoforms, regulation, disease association and outstanding questions in the field.
The figures presented are clear, useful and complement the text nicely.
I have only small suggestions/edits:
- The entire manuscript should be reviewed for gene convention. Human gene in caps and italics etc. Convention changes through the manuscript and mouse convention is frequently used for human genes/proteins and vice versa.
- Greek symbols are missing throughout the text.
- Figure 3 legend - should this be 2000 bp or 2 kbp instead of 2000 kbp?
- Line 283 - I believe TSA is a histone deacetylase inhibitor, not a histone acetylase inhibitor.
- Line 667 - typo: "varisou"
Author Response
At first, we are grateful to the reviewers for their time and constructive comments on our manuscript. We have implemented their comments and suggestions in the revised version of the manuscript. New additions to the initial version of the manuscript are highlighted in yellow in the revised version of the manuscript. Below, we also provide a point-by-point response (in blue) on how we have addressed each of the reviewer’s comments and questions.
We hope the reviewers will find our respond satisfactory and the revised manuscript suitable for publication in Genes.
Reviewer 2:
1.The entire manuscript should be reviewed for gene convention. Human gene in caps and italics etc. Convention changes through the manuscript and mouse convention is frequently used for human genes/proteins and vice versa.
Re: We thank the reviewer for the suggestion. We have edited the genes and proteins nomenclature in the text according to HUGO Gene Nomenclature Committee. Currently, human genes and transcripts are in upper-case italic (SBSN), whereas mouse genes are in 1st letter upper-case and italic (Sbsn). Human proteins are Upper case (SBSN), whereas mouse or rat proteins are 1st letter upper-case, remaining are lower-case (Sbsn). We have not highlighted these changes due to the abundance.
- Greek symbols are missing throughout the text.
Re: We are grateful for this comment. Yes, indeed, multiple Greek symbols are missing in the manuscript. This might be due to text style conversion. We have changed the Greek symbols in the text for corresponding words. Additionally, we asked the editorial office for changing the figure 4 for newly uploaded figure 4 containing adequate changes in the nomenclature.
- Figure 3 legend - should this be 2000 bp or 2 kbp instead of 2000 kbp?
Re: Thank you for the notice. We have modified the text, 2 kbp is correct.
- Line 283 - I believe TSA is a histone deacetylase inhibitor, not a histone acetylase inhibitor.
Re: We thank for pointing out the incorrect information. Indeed, TSA is a deacetylace inhibitor. The text has been updated accordingly.
- Line 667 - typo: "varisou"
Re: The typo has been corrected. Thank you.
